# A Plant-Based Diet Alleviates Molecular Pulmonary Abnormalities in Hypertension

**DOI:** 10.3390/arm93060049

**Published:** 2025-11-04

**Authors:** Rami Salim Najjar, Jaishree Jagirdar, Andrew T. Gewirtz

**Affiliations:** 1Institute for Biomedical Sciences, Georgia State University, Atlanta, GA 30303, USA; 2Division of Cardiology, Department of Medicine, Emory University School of Medicine, Atlanta, GA 30307, USA; 3Department of Pathology and Laboratory Medicine, Department of Medicine, Emory University, Atlanta, GA 30322, USA

**Keywords:** lung disease, nutrition, plant-based diet, polyphenols

## Abstract

**Highlights:**

**What are the main findings?**

**What are the implications of the main findings?**

**Abstract:**

**Background**: Essential hypertension is associated with an increased risk of pulmonary hypertension (PH). PH is diagnosed more frequently in females. Little is known about the effects of a plant-based diet (PBD) in improving lung abnormalities in PH. **Methods**: We compared 28- and 40-week-old female normotensive Wistar Kyoto and spontaneously hypertensive rats (SHR), maintained from the age of 4 weeks on a control refined diet or a PBD, comprising 28% fruits, vegetables, nuts and legumes. A subset of control SHRs were switched to the PBD at 28 weeks of age. Lungs were taken for protein and histological analysis. **Results**: Relative to WKYs, SHRs consuming the control diet exhibited decreased lung endothelial nitric oxide synthase (eNOS). PBD consumption by SHRs prevented and reversed this phenotype. Expression of E-cadherin was also reduced in SHRs. This reduction was attenuated by PBD consumption treatment. The phosphorylation of extracellular signal-regulated kinase (ERK)1/2 in the lung was increased in SHRs and attenuated by PBD. The expression of activated transforming growth factor (TGF)-β1 was also attenuated by a PBD. **Conclusions**: The PBD favorably mediated hypertension-induced pulmonary molecular abnormalities in lung endothelium, epithelial junction and pro-fibrotic signaling. Future studies should assess the effects of a PBD in improving PH and lung function.

## 1. Introduction

Pulmonary hypertension (PH) results in a 21% increased risk of mortality within 3 years of diagnosis [1]. While PH can develop in essential hypertension due to left ventricular (LV) dysfunction, in those with normal LV function, essential hypertension increases the risk of PH development by 300% [2]. Moreover, sex plays a role, as PH afflicts a greater number of females, who are 1.8–3.6 times more likely to be diagnosed compared to males [3]. PH was initially thought to be a disease afflicting mostly younger women (mean age 35); however, recent evidence suggests that it afflicts older adults as well, with 64% of new diagnoses occurring in those who are >65 years of age [4].

PH can lead to worsening lung function [5], in part due to microvascular endothelial dysfunction within lung tissue [6], resulting in poor gas exchange. This vascular stress can increase transforming growth factor (TGF)-β1, reducing epithelial adherens junction proteins, such as E-cadherin, within the lung, which increases permeability, while also promoting mesenchymal transition, as evidenced by increased epithelial expression of α-smooth muscle actin (SMA) [7]. These abnormal cellular alterations can cause congestion, scarring and overt lung disease over time. Inflammation and oxidative stress play a large part in driving these abnormal molecular processes [8,9], especially in downregulating endothelial function [10]. Targeting these molecular pathways may be of relevance in the management of lung disease induced by PH.

While studies are limited, nutrition can play a role in improving PH [11]. In particular, plant-based diets (PBDs) may be beneficial. Indeed, plant-based dietary patterns are associated with preserved lung function [12]; a PBD improves endothelial function [13] and a PBD can treat essential hypertension [14]. This may in part be due to the bioactive properties of polyphenols, secondary metabolites produced exclusively by plants, which can target the underlying pathways of inflammation, oxidative stress and endothelial dysfunction [15,16,17]. Thus, the objective of this study was to assess whether a PBD could improve lung abnormalities at the molecular level, which relates to PH as part of a secondary analysis in a model of essential hypertension.

## 2. Materials and Methods

This investigation is a secondary analysis of an existing study which evaluated coronary microvascular dysfunction [18]. This animal study was approved by Georgia State University’s Institutional Animal Care and Use Committee (protocol #: A23025). All animals were euthanized by CO_2_ affixation, followed by decapitation at either week 24 or week 36. PH is more prevalent in women compared to men; thus, females are appropriate to use in this model. Spontaneously hypertensive rats (SHRs), while a model of essential hypertension, have been validated to develop PH by 14–18 weeks of age, along with subsequent lung histopathological abnormalities [19]. Thus, SHRs are appropriate models of PH.

Wistar Kyoto (WKY) rats served as normotensive controls. All animals were female and three weeks old upon arrival. Further, animals were purchased from Inotiv (West Lafayette, IN, USA), within their own in-bred colonies. Upon arrival, rats were housed in pairs in an environmentally controlled animal care facility (50 ± 5% relative humidity, 20–25 °C) and maintained on 12 h light/dark cycles. Rats were doubly housed, since single animals may have altered hemodynamics due to increased stress [20]. Rats were maintained on a purified control diet (Appendix A) [18]. After one week, SHRs either continued the control diet or were switched to a PBD. WKYs consumed the control diet for the entirety of the study. The PBD comprised 28% (*w*/*w*) of seven different plant foods: walnuts, black beans, red bell pepper, sweet potato, blueberries, brussels sprouts and lemon (4% each). The diversity of foods was to better reflect a diverse human PBD. The selection of these foods was based on two criteria: (1) high polyphenol content [21] and (2) commonly consumed in the United States. While the control diet contained casein, the PBD used soy protein instead (Appendix A). The control diet and PBD were nearly identically matched in nutritional composition, including protein, carbohydrates, fat and fiber (insoluble and soluble), as well as vitamins and minerals. As such, the main known difference between the PBD and control diet was the polyphenol content. Polyphenol intake in rats consuming a PBD was estimated to be ~2582 mg/kg BW, based on average food intake, body weight of animals and available polyphenol analysis data from Phenol Explorer [22,23]. This corresponds to ~96 mg/kg BW of polyphenols in human equivalents [24], or 5760 mg of polyphenols per 2000 Kcal. The control diet did not contain any whole plant foods, and can be considered a proxy for a Western dietary pattern, which lacks plant foods [25].

The prevention phase comprised animals consuming their respective diets for 24 weeks, starting from 4 weeks of age (Figure 1). For additional clinical relevance, we also utilized a treatment model, in which a subset of SHRs on the control diet that were hypertensive for 24 weeks were switched to a PBD for 12 additional weeks. Otherwise, rats continued the control diet for 12 weeks during the treatment phase. Figure 1 outlines the animal study design. At the time of sacrifice, SHRs, irrespective of diet, remained hypertensive (systolic BP: ~160 mmHg), while WKYs were normotensive (systolic BP: ~112 mmHg) [18].

### 2.1. Western Blot

The lung was excised, immediately frozen in liquid nitrogen, and stored at −80 °C for downstream protein analysis. Lung tissue (20–40) mg was homogenized in 400 µL RIPA (with protease and phosphatase inhibitors), using a glass Dounce homogenizer. Protein lysates from the tissue were centrifuged at 16,000× *g* for 20 min and supernatants were collected. The protein concentration of lysates was determined using the DC protein assay kit (BioRad Laboratories, Hercules, CA, USA). For Western blot, 60 µg of protein from tissue were separated in 8–15% SDS-PAGE gels (supplemented with 2-2-2-trichloroethanol, with a final concentration of 0.5% for total lane protein visualization) and transferred to polyvinylidene difluoride (PVDF) membranes, using Trans-Blot Turbo (BioRad Laboratories). Enhanced chemiluminescence (WBLUF0500, Millipore Sigma, Darmstadt, Germany) was used to determine the expression of proteins. The density of protein bands was quantified using Image Lab 6.0 (BioRad Laboratories, Hercules, CA, USA), which was normalized to total lane protein. Phosphorylated proteins were normalized to their respective total protein counterparts. If phosphorylated and total proteins were on different membranes, then each protein was first normalized to the total lane protein. Antibodies and their respective catalog numbers are listed in Table 1.

### 2.2. Histology

At sacrifice, lungs were stored in 10% formalin for 24 h at 4° C. After this, formalin was replaced with 70% ethanol and stored at room temperature. Tissue samples were dehydrated and embedded in paraffin for histological analysis. Cuts were made at 10 µm thickness. Morphology and fibrosis were assessed by hematoxylin and eosin (H&E) staining and Sirius red staining, respectively.

### 2.3. Statistical Analysis

GraphPad Prism (v10.6; San Diego, CA, USA) was used for all statistical analyses. Normality was assessed with a Shapiro–Wilk test, and all data were normally distributed. Pairwise comparisons were made using Student’s *t*-test between WKY and SHR or SHR and SHR + PBD. Values are represented as mean ± standard deviation (SD). Data were deemed significant if *p* ≤ 0.05. Raw *p*-values were reported in figures if *p*-value < 0.1.

## 3. Results

### 3.1. Impacts of a Plant-Based Diet on Lung Endothelial Nitric Oxide Synthase (eNOS) in Hypertension

As an indirect means of probing endothelial dysfunction, we measured the expression of endothelial nitric oxide synthase (eNOS), a nitric oxide-producing enzyme whose levels are associated with endothelial dysfunction [26,27,28]. The expression of endothelial nitric oxide synthase (eNOS) was significantly reduced in SHRs consuming the control diet relative to WKYs at week 24 (Figure 2A,B; 0.74 ± 0.09 vs. 1.00 ± 0.22), which progressively worsened by week 36 (Figure 2C,D; 0.56 ± 0.041 vs. 1.00 ± 0.17). A plant-based diet prevented this decline in eNOS (Figure 2A,B; 1.04 ± 0.12) and reversed it (Figure 2C,D; 0.94 ± 0.20).

### 3.2. Impacts of a Plant-Based Diet on Expression of Proteins Involved in Lung Epithelial Integrity and Fibrosis in Hypertension

Contrary to our hypothesis, there was no significant increase in vimentin expression in the lungs of SHRs consuming the control diet at weeks 24 (Figure 3A,B) and 36 (Figure 3E,F) compared to WKYs, suggesting that there was not a mesenchymal transition of epithelial cells. Nonetheless, E-cadherin, an adherens junction protein involved in epithelial integrity, was significantly reduced in SHRs consuming the control diet relative to WKYs at week 24 (Figure 3A,C, 0.82 ± 0.10 vs. 1.00 ± 0.18), which worsened at week 36 (Figure 3E,G; 0.75 ± 0.08 vs. 1.00 ± 0.11). While PBD supplementation did not improve E-cadherin during the prevention phase at week 24 (0.76 ± 0.05, *p* = 0.15), it did attenuate its decline during the treatment phase at week 36 (0.86 ± 0.08). The phosphorylation of extracellular signal-regulated kinase (ERK)1/2, which is closely tied to the loss of E-cadherin [29], was not significantly different between WKYs and SHRs (*p* = 0.22); however, its phosphorylation was reduced in SHR + PBD vs. SHR alone (Figure 3A,D, 0.81 ± 0.12 vs. 1.19 ± 0.42). At week 36, however, ERK1/2 phosphorylation continued to increase (Figure 3E,H) in SHRs consuming the control diet (1.74 ± 0.54), which was significantly greater than WKYs (1.00 ± 0.22) and SHR + PBD (1.18 ± 0.34).

The expression of matrix metalloproteinase (MMP)9, a protein typically increased in lung tissue in PH [30], was unchanged between animals in any experimental group (Figure 4A,B,G,H). The expression of α-SMA, a protein expressed in epithelial cells during mesenchymal transition and in activated fibroblasts, was significantly increased in SHRs vs. WKY at week 24 (Figure 4A,C, 1.46 ± 0.22 vs. 1.00 ± 0.13) and was attenuated by a PBD (1.11 ± 0.23). However, no significant differences in α-SMA were observed at week 36 between groups (Figure 4G,I), although a nonsignificant increase in SHR + PBD was observed vs. SHRs alone. TGF-β1, a protein involved in reducing E-cadherin, mesenchymal transition and promoting fibrosis, was significantly impacted by the presence of hypertension. The uncleaved and inactive form, pro-TGF-β1, was reduced in SHRs consuming the control diet vs. WKYs at week 24 (Figure 4A,D, 0.77 ± 0.32 vs. 1.00 ± 0.06), albeit not significantly (*p* = 0.0843), which was reduced to an even greater extent at week 36 (Figure 4G,J, 0.55 ± 0.13 vs. 1.00 ± 0.09) and significantly so (*p* = 0.0001). A PBD was as effective in WKY at attenuating pro-TGF-β1 cleavage vs. SHR alone. A reduction in uncleaved TGF-β1 implies that there is more cleaved and active TGF-β1. Indeed, at week 24 (Figure 4A,E), a significant increase in cleaved TGF-β1 was observed in SHRs alone vs. WKY (1.72 ± 0.49 vs. 1.00 ± 0.10). While a PBD did not significantly reduce the expression of cleaved TGF-β1 (1.41 ± 0.12, *p* = 0.11), the ratio of cleaved/uncleaved TGF-β1 (Figure 4A,F), an indicator of how much relative TGF-β1 that is produced is in its active form, was significantly reduced in SHR + PBD vs. SHR alone (1.86 ± 0.33 vs. 2.28 ± 0.34, *p* = 0.0401) and in WKY vs. SHR (1.00 ± 0.10, *p* <0.0001). At week 36, there were no significant differences between groups in the expression of cleaved TGF-β1 (Figure 4G,K); however, the ratio of cleaved/uncleaved TGF-β1 was significantly increased in SHRs consuming the control diet relative to WKY (Figure 4G,L, 2.06 ± 0.62 vs. 1.00 ± 0.42, *p* = 0.0069) and relative to SHR + PBD (1.31 ± 0.20, *p* = 0.0170).

### 3.3. Impacts of a Plant-Based Diet on Lung Inflammatory Signaling Proteins in Hypertension

No significant differences in the immune cell infiltration marker, F4/80, were observed between groups at week 24 (Figure 5A,B) or week 36 (Figure 5G,H); nor was the phosphorylation of NF-κB significantly impacted (Figure 5A,C,G,I). However, the phosphorylation of p38, a mitogen-activated protein kinase (MAPK) involved in vascular remodeling [31], was increased in SHRs consuming the control diet vs. WKYs (1.86 ± 0.88 vs. 1.00 ± 0.54, *p* = 0.0504) and SHR + PBD vs. SHR alone (3.50 ± 1.2, *p* = 0.0210) at week 24 (Figure 5A,D). At week 36 (Figure 5G,J), differences in phospho-p38 between SHR and SHR + PBD were no longer significant (*p* = 0.2597); however, the expression of this protein remained elevated in SHRs consuming the control diet vs. WKYs (2.08 ± 0.69 vs. 1.00 ± 0.64, *p =* 0.0170). At week 24, the expression of phosphorylated stress-activated protein kinases (SAPK)/Jun amino-terminal kinases (JNK) (Figure 5A,E), a protein also involved in vascular remodeling [32], was increased (non-significantly) in SHRs consuming the control diet vs. WKYs (1.65 ± 0.82 vs. 1.00 ± 0.27, *p* = 0.0651) and vs. PBD (1.10 ± 0.30, *p* = 0.0980). This change, however, was significant in the phosphorylation of c-Jun (Figure 5A,F), a transcription factor directly activated by SAPK/JNK at week 24. No significant differences were observed in the phosphorylation of SAPK/JNK and c-Jun at week 36 (Figure 5G,K,L).

### 3.4. Impacts of a Plant-Based Diet in Pulmonary Redox Proteins

The expression of xanthine oxidase (XO), a producer of superoxide, was significantly increased at week 24 (Figure 6A,B) in WKY vs. SHR (1.00 ± 0.17 vs. 0.67 ± 0.25, *p* = 0.0227) and SHR vs. SHR + PBD (0.67 ± 0.25 vs. 0.44 ± 0.05, *p* = 0.0419). At week 36, the expression of XO (Figure 6F,G) was no longer significantly different between WKY and SHR (*p* = 0.25), while PBD supplementation non-significantly reduced XO expression vs. SHR alone (1.40 ± 1.11 vs. 0.55 ± 0.38, *p* = 0.0732). The expression of the NADPH-oxidase 2 subunit, p47phox, involved in the production of superoxide, was significantly reduced at week 24 (Figure 6A,C) and week 36 (Figure 6F,H) in SHRs consuming the control diet vs. WKYs, with no effect of PBD supplementation. The expression of the NADPH-oxidase subunit p22phox, which is ubiquitous among NADPH-oxidase isoforms, was not changed between groups at week 24 (Figure 6A,D), but was significantly increased in SHRs consuming the control diet vs. WKYs (1.17± 0.05 vs. 1.00 ± 0.05, *p* = 0.0185) at week 36 (Figure 6F,I). As a proxy for oxidative stress, the expression of 3-nitrotyrosine (3-NT) was not changed between WKY and SHR at week 24 (Figure 6A,E). In contrast, PBD supplementation significantly reduced 3-NT expression vs. SHRs consuming the control diet (0.54 ± 0.05 vs. 0.95 ± 0.28, *p* = 0.0068). At week 36, however, during the treatment phase, the expression of 3-NT (Figure 6F,J) was non-significantly reduced in SHRs consuming the control diet vs. WKYs (0.88 ± 0.12 vs. 1.00 ± 0.10, *p* = 0.0787), with no effect of PBD supplementation observed (*p* = 0.1369).

The expression of the antioxidant enzyme, superoxide dismutase (SOD)1, which neutralizes superoxide in the cytosol, was not significantly different between groups at week 24 (Figure 7A,B) or week 36 (Figure 7G,H). However, SOD2, a mitochondrial superoxide-neutralizing protein, was significantly reduced at week 24 (Figure 7A,C) in SHRs consuming the control diet vs. WKYs (0.75 ± 0.18 vs. 1.00 ± 0.13, *p* = 0.0007), which reduced further at week 36 (Figure 7G,I) in SHR, relative to WKY (0.67 ± 0.08 vs. 1.00 ± 0.06, *p* < 0.0001). PBD did not significantly impact the expression of SOD2 at any time point. The antioxidant enzyme catalase (CAT), a protein that converts hydrogen peroxide to water, was significantly reduced in SHRs consuming the control diet vs. WKYs (0.64 ± 0.14 vs. 1.00 ± 0.0.25, *p* = 0.0138) at week 24 (Figure 7A,D), for which PBD supplementation attenuated (0.78 ± 0.11, *p* = 0.0672). Differences in CAT were no longer statistically significant between groups at week 36 (Figure 7G,J). Glutathione peroxidase (GPx)1, an enzyme that also neutralizes hydrogen peroxide, was significantly reduced at week 24 (Figure 7A,E) in SHRs consuming the control diet vs. WKYs (0.52 ± 0.13 vs. 2.00 ± 0.08, *p* < 0.0001). PBD supplementation did not significantly change GPx1 expression relative to SHRs alone (*p* = 0.2045). No significant differences in GPx1 expression were observed between groups at week 36 (Figure 7G,K). The expression of nuclear factor erythroid 2-related factor 2 (NRF2), a transcription factor and regulator of glutathiones, was reduced (non-significantly) at week 24 (Figure 7A,F) in SHRs consuming the control diet vs. WKYs (0.67 ± 0.19 vs. 1.00 ± 0.35, *p* = 0.0519). PBD supplementation did not significantly impact the expression of NRF2 relative to HSR (*p* = 0.1314). At week 36, however, there were no significant differences in the expression of NRF2 between groups (Figure 7 G,L).

### 3.5. Impacts of PBD on Lung Architecture and Fibrosis in Female SHRs

No pathological alterations in lung architecture or blood vessels were observed between WKY and SHR, with similar observations found in SHR + PBD (Figure 8A). Similarly, no significant differences in collagen were observed via Sirius red staining (Figure 8B).

## 4. Discussion

A plant-based diet both prevented and treated pathological molecular changes in proteins involved in endothelial function, epithelial junction and fibrosis in the lung (Figure 9). However, despite these changes, no changes in lung architecture or differences in collagen deposition were noted. This is in contrast to previous findings, in which 14–18 week-old male SHRs had significant pathological alteration in lung tissue [19]. We used female rats that were 28 or 40 weeks old. However, female rats have rarely been utilized in studies of PH; thus, we reveal a key difference in phenotype between sexes. It is possible, however, that with aging, more pronounced pathological changes in lung architecture would be observed. For example, male SHRs tend to develop heart failure 6–12 months sooner than female SHRs [33,34]. Nonetheless, human PH is more dominant in women; thus, female SHRs are more translational than male SHRs.

Changes in inflammatory signaling and redox proteins were less clear and more ambiguous (Figure 5, Figure 6 and Figure 7), and conclusions are difficult to draw from this data as a result. While a clear reduction in XO in SHRs consuming PBD versus SHRs consuming the control diet was observed, XO was lower in SHRs at week 24 vs. WKYs, but not at week 36, while 3-NT was lower in PBD vs. SHR at week 24, but not week 36. Despite these discrepancies, a clear finding we observed was complete prevention and reversal of diminished eNOS protein expression in lung tissue with PBD supplementation (Figure 2). This suggests that a PBD could improve lung microvascular endothelial function independently of hypertension. This is important, as microvascular dysfunction of lungs leads to poor gas exchange, potential hypoxia and lung injury, which drives the development of PH [6]. As part of these pathological changes in SHRs, E-cadherin was downregulated, which a PBD attenuated (Figure 3), and activated TGF-β1 was increased, which a PBD also attenuated during both the prevention and treatment phase (Figure 4). Considering these molecular changes in the lungs of SHRs and the protective action of PBD, it is likely that a PBD could prevent and/or treat PH, although additional studies are needed. Because this study is a secondary analysis, pulmonary hypertension was not directly assessed: nor was lung function.

In humans, polyphenol-rich fruit intake is associated with improved lung function [35,36]. Additionally, healthy PBDs, characterized by the intake of whole fruits, vegetables, whole grains and legumes, were associated with a 46% reduced risk of chronic obstructive pulmonary disease (COPD), while an unhealthy PBD, characterized by refined grains and processed foods, was associated with a 39% higher risk [37]. Even in the presence of air pollution, a healthy PBD can counteract the increased risk of COPD, while an unhealthy PBD exacerbates it [38]. This is likely due to the polyphenol and phytochemical content of healthy PBDs, as whole plant foods are rich sources of polyphenols compared to their refined counterparts. Indeed, polyphenol-rich foods and other isolated phytochemicals have been found to be efficacious in animal models of PH [39]. In a model of hypoxia-induced PH, for example, apple polyphenols improved both endothelial-dependent and vascular smooth muscle cell-dependent function in isolated pulmonary arteries [40]. The expression of eNOS also increased. In the lungs specifically, blueberry extract was found to improve lung function and reduce lung oxidative stress in an animal model of monocrotaline-induced PH, using male rats [41]. In this model, however, NRF2 and SODs were upregulated with the blueberry extract, which was not observed in the present study (Figure 7), despite blueberries being a part of the PBD. The mix of plant foods in the PBD used in this study was used in an effort to partially recapitulate a human PBD; thus, our findings are more translational than food extracts or single phytochemical interventions.

Despite the protective role of estrogen in cardiovascular diseases [42], females, including those that are premenopausal, have higher rates of PH than males. Oddly, estrogen is protective in PH in animal models [43,44], representing a paradox. Thus, it is not clear why females have a higher incidence of PH. Future studies should evaluate sex differences in SHRs and also evaluate young versus aged animals to elucidate these mechanisms.

Several limitations exist in the present study. Firstly, we did not use a traditional model of PH [45]. Instead, we used an essential hypertension model in the SHR. However, the existing literature illustrates that SHRs develop PH [19,46,47,48,49]. Thus, we have confidence that the model is appropriate. Secondly, we did not directly measure PH in these animals, which may have been attenuated by the intervention. However, because systemic hypertension was not impacted by PBD supplementation relative to SHRs consuming the control diet, it is likely that PH was unchanged in SHRs between diets; however, future studies should validate this. In addition, we cannot be certain whether the majority of eNOS is from endothelial cells or from other cell types in the lungs which also express eNOS. Prior studies have found that eNOS activity from lung endothelial cells in PH is downregulated—thus, we believe that our data reflects the microvascular endothelium, at least partially [50]. Lung epithelial cells also express eNOS, but this expression is half of that of lung endothelial cells [51]. We also are not certain whether our junction markers (Figure 3) reflect purely epithelial cells. However, we aimed to specifically collect lung tissue for protein analysis at the terminal end of the tissue, rather than towards the large bronchi; as such, we believe we have captured protein from the end of the bronchial tree which comprises the small airway and contains substantial alveoli. In addition, E-cadherin is observed primarily in epithelial cells of the small airways [52]. As such, we believe that our data primarily reflects epithelial junction proteins, rather than junction proteins of other cell types. Lastly, while we did not observe histopathological changes in the lung, the known pathological effects of the changes in the expression of proteins involved in endothelial function, epithelial junctions and fibrosis were attenuated, suggesting that female SHRs of an increased age would likely be needed to show histological abnormalities.

## 5. Conclusions

In conclusion, a PBD favorably improved proteins associated with PH in the lung, using both prevention and treatment models in female SHRs. This reveals a potential novel therapeutic strategy in the treatment of lung abnormalities associated with PH. Future studies should make direct comparisons between sexes and assess whether a PBD improves PH. Pilot clinical studies utilizing a PBD to treat human PH may be warranted; however, confirmation of the ability of a PBD to target PH itself requires investigation.

## Figures and Tables

**Figure 1 arm-93-00049-f001:**
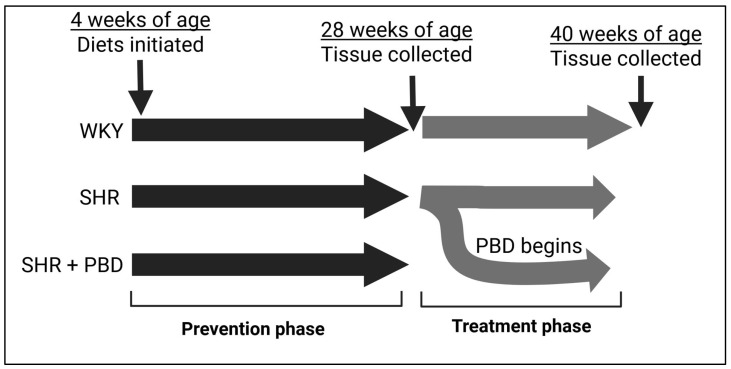
Study design. Abbreviations: PBD, plant-based diet; SHR, spontaneously hypertensive rat and WKY, Wistar Kyoto.

**Figure 2 arm-93-00049-f002:**
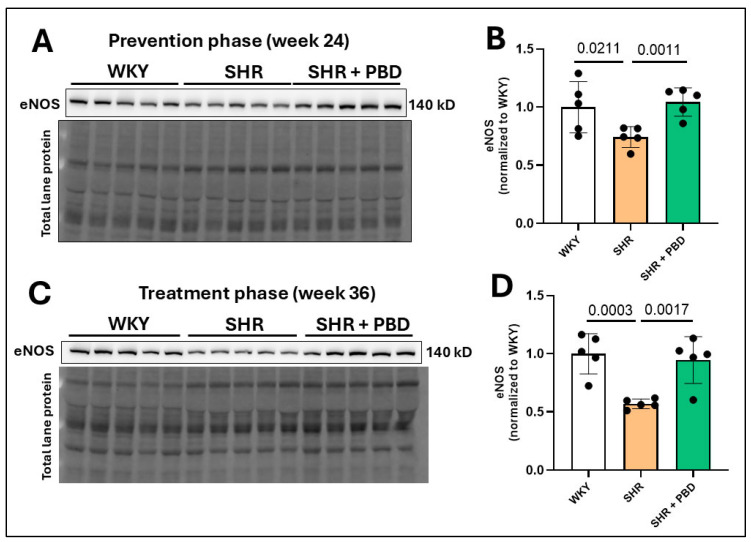
Lung microvascular eNOS protein expression. At 4 weeks of age, spontaneously hypertensive rats (SHRs) consumed either a control purified diet or a plant-based diet (PBD) for 24 weeks. Wistar Kyoto (WKY) rats also consumed the control diet for 24 weeks. Animals were then euthanized (prevention phase). For the treatment phase, a subgroup of SHRs were switched from the control diet to a PBD after 24 weeks, while the remaining WKYs and SHRs continued the control diet for 12 additional weeks. Lung protein was isolated, and eNOS was assessed at (**A**,**B**) week 24 or (**C**,**D**) week 36 via Western blot. Statistical comparisons were made with Student’s *t*-test between WKY vs. SHR or SHR vs. SHR + PBD (*n* = 5/group). Data are expressed as mean ± SD. Abbreviations: eNOS, endothelial nitric oxide synthase; PBD, plant-based diet; SHR, spontaneously hypertensive rat and WKY, Wistar Kyoto.

**Figure 3 arm-93-00049-f003:**
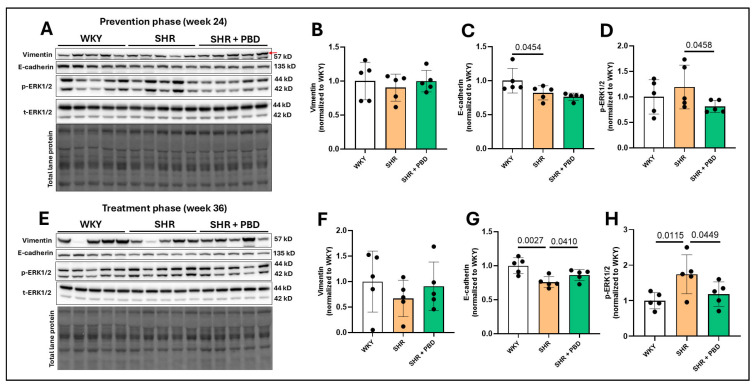
Expression of mesenchymal transition and junction proteins. At 4 weeks of age, spontaneously hypertensive rats (SHRs) consumed either a control purified diet or a plant-based diet (PBD) for 24 weeks. Wistar Kyoto (WKY) rats also consumed the control diet for 24 weeks. Animals were then euthanized (prevention phase). For the treatment phase, a subgroup of SHRs were switched from the control diet to a PBD after 24 weeks, while the remaining WKYs and SHRs continued the control diet for 12 additional weeks. Lung protein was isolated and protein expression of (**A**,**B**,**E**,**F**) Vimentin, (**A**,**C**,**E**,**G**) E-cadherin and (**A**,**D**,**E**,**H**) phospho-ERK1/2 was assessed at (**A**–**D**) week 24 or (**E**–**H**) week 36, via Western blot. The red arrow in the vimentin blot for panel A points to the band of interest. Statistical comparisons were made with Student’s *t*-test between WKY vs. SHR or SHR vs. SHR + PBD (*n* = 5/group). Data are expressed as mean ± SD. Abbreviations: ERK, extracellular signal-regulated kinases; PBD, plant-based diet; SHR, spontaneously hypertensive rat and WKY, Wistar Kyoto.

**Figure 4 arm-93-00049-f004:**
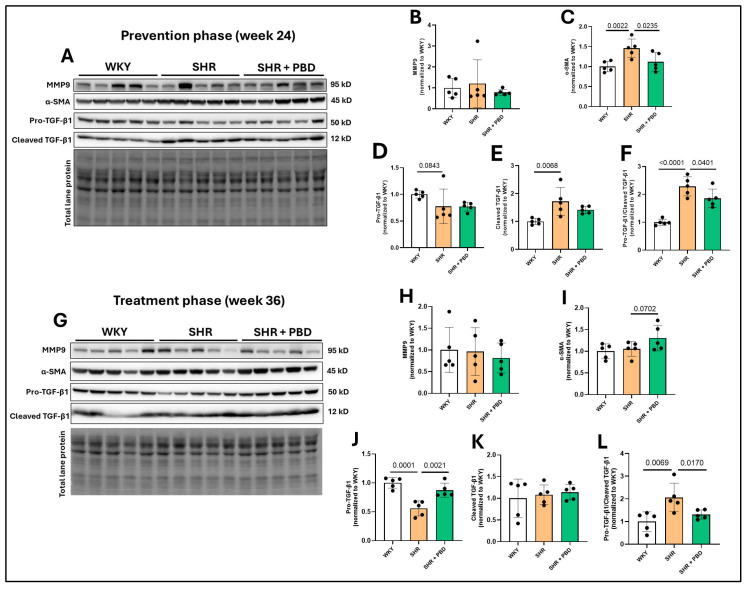
Expression of proteins involved in fibrosis. At 4 weeks of age, spontaneously hypertensive rats (SHRs) consumed either a control purified diet or a plant-based diet (PBD) for 24 weeks. Wistar Kyoto (WKY) rats also consumed the control diet for 24 weeks. Animals were then euthanized (prevention phase). For the treatment phase, a subgroup of SHRs were switched from the control diet to a PBD after 24 weeks, while the remaining WKYs and SHRs continued the control diet for 12 additional weeks. Lung protein was isolated and protein expression of (**A**,**B**,**G**,**H**) MMP9, (**A**,**C**,**G**,**I**) α-SMA, (**A**,**D**,**G**,**J**) pro-TGF-β1, (**A**,**E**,**G**,**K**) cleaved TGF-β1 and (**A**,**F**,**G**,**L**) the ratio of cleaved/pro-TGF-β1, at (**A**–**F**) week 24 or (**G**–**L**) week 36 via Western blot. Statistical comparisons were made with Student’s *t*-test between WKY vs. SHR or SHR vs. SHR + PBD (*n* = 5/group). Data are expressed as mean ± SD. Abbreviations: MMP, matrix metalloproteinase; PBD, plant-based diet; SHR, spontaneously hypertensive rat; SMA, smooth muscle actin; TGF, transforming growth factor; WKY, Wistar Kyoto.

**Figure 5 arm-93-00049-f005:**
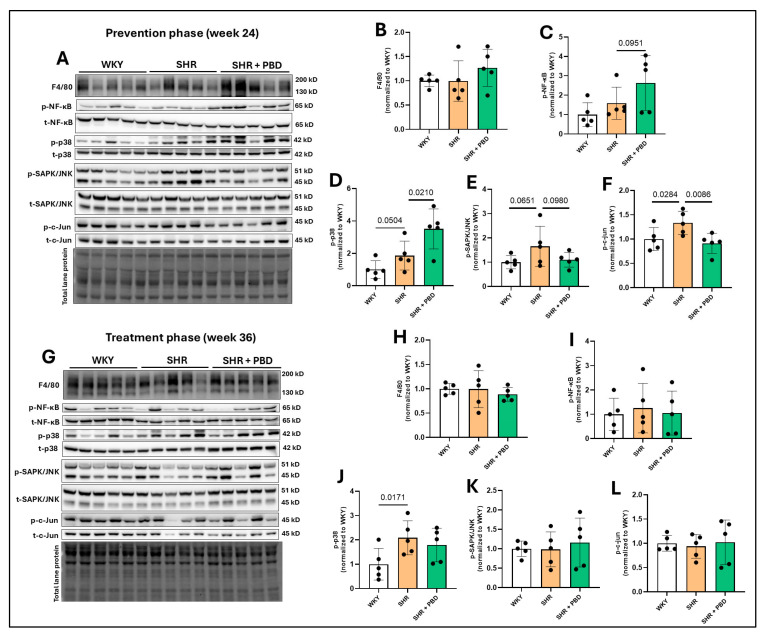
Expression of inflammatory proteins. At 4 weeks of age, spontaneously hypertensive rats (SHRs) consumed either a control purified diet or a plant-based diet (PBD) for 24 weeks. Wistar Kyoto (WKY) rats also consumed the control diet for 24 weeks. Animals were then euthanized (prevention phase). For the treatment phase, a subgroup of SHRs were switched from the control diet to a PBD after 24 weeks, while the remaining WKY and SHRs continued the control diet for 12 additional weeks. Lung protein was isolated and protein expression of (**A**,**B**,**G**,**H**) F4/80, (**A**,**C**,**G**,**I**) p-NF-κB, (**A**,**D**,**G**,**J**) p-p38, (**A**,**E**,**G**,**K**) p-SAPK/JNK and (**A**,**F**,**G**,**L**) p-c-Jun, at (**A**–**F**) week 24 or (**G**–**L**) week 36 via Western blot. Statistical comparisons were made with Student’s *t*-test between WKY vs. SHR or SHR vs. SHR + PBD (*n* = 5/group). Data are expressed as mean ± SD. Abbreviations: NF-κB, nuclear Factor kappa-light-chain-enhancer of activated B cells; PBD, plant-based diet; SAPK/JNK, stress-activated protein kinases/Jun amino-terminal kinases; SHR, spontaneously hypertensive rat and WKY, Wistar Kyoto.

**Figure 6 arm-93-00049-f006:**
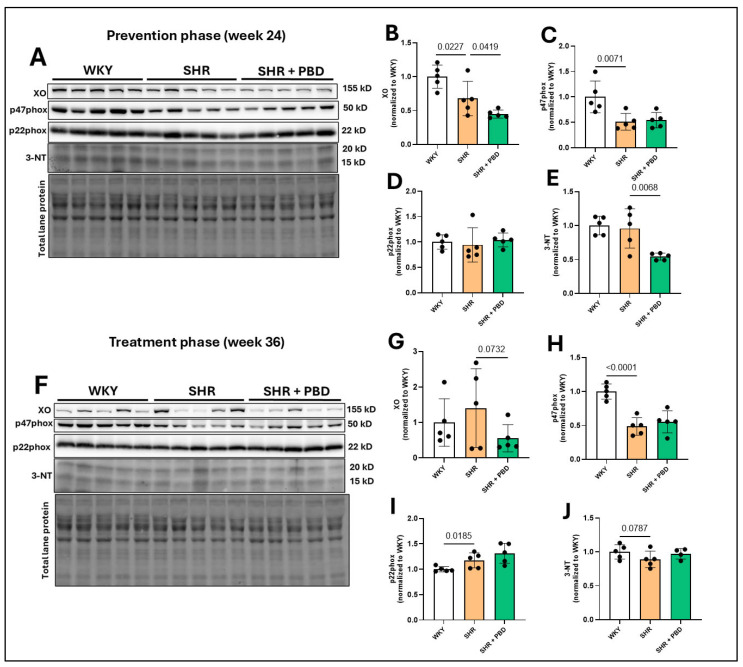
Expression of pro-oxidative enzymes. At 4 weeks of age, spontaneously hypertensive rats (SHRs) consumed either a control purified diet or a plant-based diet (PBD) for 24 weeks. Wistar Kyoto (WKY) rats also consumed the control diet for 24 weeks. Animals were then euthanized (prevention phase). For the treatment phase, a subgroup of SHRs were switched from the control diet to a PBD after 24 weeks, while remaining WKY and SHRs continued the control diet for 12 additional weeks. Lung protein was isolated and protein expression of (**A**,**B**,**F**,**G**) XO, (**A**,**C**,**F**,**H**) p47phox, (**A**,**D**,**F**,**I**) p22phox and (**A**,**E**,**F**,**J**) 3-NT, at (**A**–**E**) week 24 or (**F**–**J**) week 36 via Western blot. Statistical comparisons were made with Student’s *t*-test between WKY vs. SHR or SHR vs. SHR + PBD (*n* = 5/group). Lane 13 was excluded (**F**) because of a large, vertical, nonspecific line running through the bands of interest. The raw blot can be found in the Appendix A. Data are expressed as mean ± SD. Abbreviations: NT, nitrotyrosine; PBD, plant-based diet; SHR, spontaneously hypertensive rat; WKY, Wistar Kyoto and XO, xanthine oxidase.

**Figure 7 arm-93-00049-f007:**
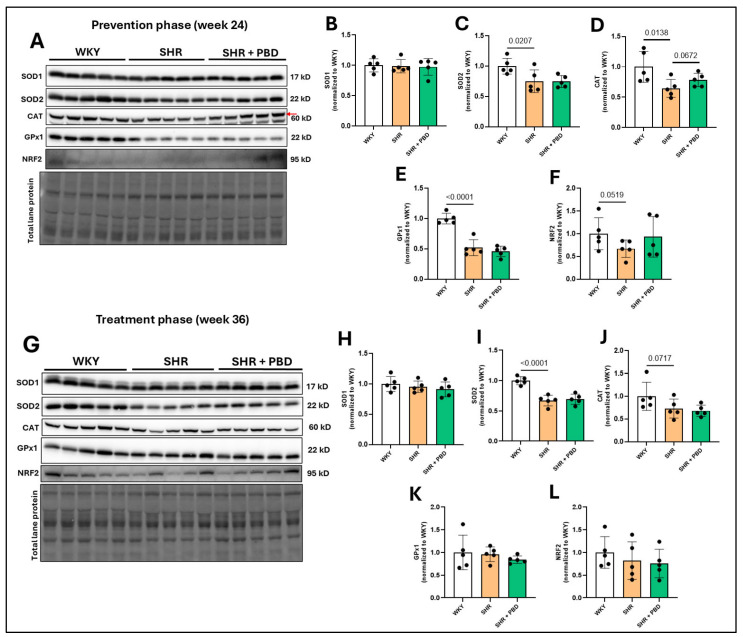
Expression of antioxidant proteins. At 4 weeks of age, spontaneously hypertensive rats (SHRs) consumed either a control purified diet or a plant-based diet (PBD) for 24 weeks. Wistar Kyoto (WKY) rats also consumed the control diet for 24 weeks. Animals were then euthanized (prevention phase). For the treatment phase, a subgroup of SHRs were switched from the control diet to a PBD after 24 weeks, while the remaining WKYs and SHRs continued the control diet for 12 additional weeks. Lung protein was isolated and protein expression of (**A**,**B**,**G**,**H**) SOD1, (**A**,**C**,**G**,**I**) SOD2, (**A**,**D**,**G**,**J**) CAT, (**A**,**E**,**G**,**K**) GPx1 and (**A**,**F**,**G**,**L**) NRF2, at (**A**–**F**) week 24 or (**G**–**L**) week 36, via Western blot. The red arrow in the CAT blot for panel A points to the band of interest. Statistical comparisons were made with Student’s *t*-test between WKY vs. SHR or SHR vs. SHR + PBD (*n* = 5/group). Data are expressed as mean ± SD. Abbreviations: CAT, catalase; GPx, glutathione peroxidase; NRF2, nuclear factor erythroid 2-related factor 2; PBD, plant-based diet; SHR, spontaneously hypertensive rat; SOD, superoxide dismutase and WKY, Wistar Kyoto.

**Figure 8 arm-93-00049-f008:**
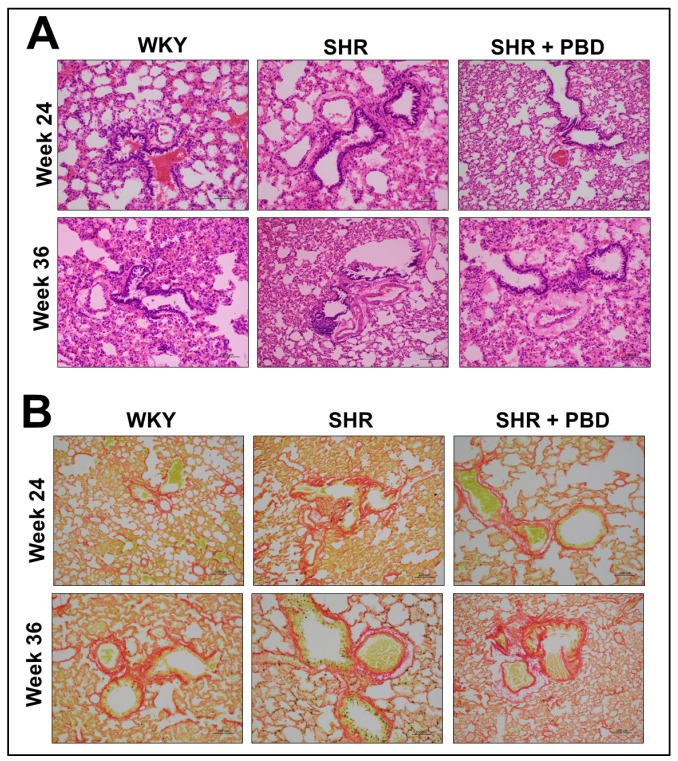
Histological staining of lung tissue sections. At 4 weeks of age, spontaneously hypertensive rats (SHRs) consumed either a control purified diet or a plant-based diet (PBD) for 24 weeks. Wistar Kyoto (WKY) rats also consumed the control diet for 24 weeks. Animals were then euthanized (prevention phase). For the treatment phase, a subgroup of SHRs were switched from the control diet to a PBD after 24 weeks, while the remaining WKYs and SHRs continued the control diet for 12 additional weeks. Lungs were excised and stained for either (**A**) H&E or (**B**) Sirius red. Red staining in panel B indicates fibrosis. Total magnification is 200×. Abbreviations: PBD, plant-based diet; SHR, spontaneously hypertensive rat and WKY, Wistar Kyoto.

**Figure 9 arm-93-00049-f009:**
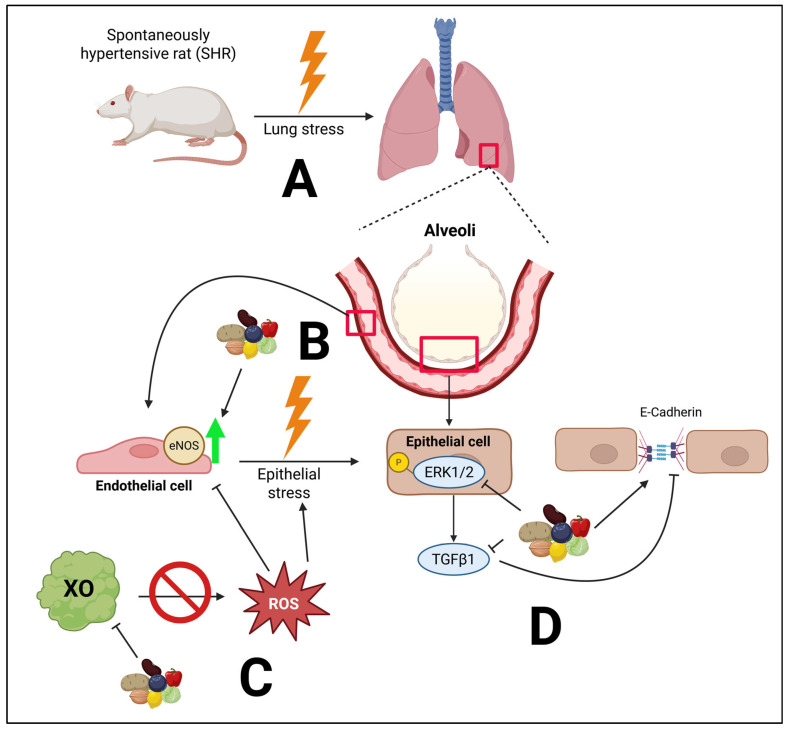
Overall hypothesized mechanisms by which a PBD favorably impacts the lungs in essential hypertension. A major cause of this stress is microvascular endothelial dysfunction. (**A**) Essential hypertension can lead to pulmonary hypertension and lung stress. (**B**) Compromised eNOS can drive epithelial stress. A PBD preserves the expression of eNOS, thus potentially improving microvascular endothelial dysfunction. (**C**) A PBD inhibits XO expression, likely reducing the release of superoxide, a form of ROS, which can compromise endothelial function and drive epithelial stress. (**D**) The phosphorylation of ERK1/2 can increase TGFβ1 activity, resulting in reduced E-cadherin, compromising barrier function. However, a PBD attenuated these molecular abnormalities. Created in BioRender. Najjar, R. (2025) https://BioRender.com/f926qw3. Biorender publication agreement number: MX28YBH0G6.

**Table 1 arm-93-00049-t001:** Antibodies used in this study for Western blot.

Protein Target	Company *	Catalog Number
3-nitrotyrosine	Cell Signaling	92212
α-SMA	Cell Signaling	19245
c-Jun	Cell Signaling	9165
phospho-c-Jun	Cell Signaling	9261
Catalase	Cell Signaling	14097
E-cadherin	Cell Signaling	3195
eNOS	Cell Signaling	32027
ERK1/2	Cell Signaling	9102
Phospho-ERK1/2	Cell Signaling	9101
GPx1	R&D	AF3798
MMP9	Cell Signaling	24317
NF-κB	Cell Signaling	4764
NRF2	Cell Signaling	20733
phospho-NF-κB	Cell Signaling	3033
p22phox	Cell Signaling	27297
p38	Cell Signaling	8690
phospho-p38	Cell Signaling	4511
p47phox	Cell Signaling	63290
SAPK/JNK	Cell Signaling	9252
phospho-SAPK/JNK	Cell Signaling	4668
SOD1	Cell Signaling	37385
SOD2	Cell Signaling	13141
TGF-β1	Abcam	ab215715
Vimentin	Cell Signaling	5741
Xanthine oxidase	Abcam	ab109235
Rabbit secondary	Cell Signaling	7074
Goat secondary	R&D	HAF109

* City, state and country of each vendor: Abcam (Waltham, MA, USA); Cell Signaling (Danvers, MA, USA) and R&D (Minneapolis, MN, USA).

## Data Availability

The original contributions presented in this study are included in the article/Appendix A. Further inquiries can be directed to the corresponding author.

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
