# Peer review of "A Plant-Based Diet Alleviates Molecular Pulmonary Abnormalities in Hypertension"

_arm, 2025, doi:10.3390/arm93060049_

Round 1
Reviewer 1 Report
Comments and Suggestions for Authors
The article is highly interesting and underscores the importance of a healthy diet in producing beneficial effects for the prevention or potential treatment of pulmonary hypertension induced by systemic hypertension in preclinical studies with SHR rats. Nevertheless, I have several questions and suggestions, presented as follows:
Line 45: Add a space after “males […]”.
Line 59-64: Please replace the references, since i think that adding different studies improve the strong of the manuscrip.
Line 69: Consider adding more information regarding the higher prevalence of PH in women compared to men—for example, age-related differences.
Line 75–76: Why were the rats doubly housed? Please also provide information regarding the diet, nutrition, and health care of the animal model prior to the experimental protocol, as well as the room temperature and whether all procedures were approved by an ethics committee.
Line 79–83: Why was the concentration of each PBD component set at 4%? Why was it the same for all components, despite their differing polyphenol contents? Finally, what quantity of PBD was administered to each rat, and why?
Lines 92–93: Add details regarding the methods used for euthanasia.
Figure 1: Please include explanations for all acronyms in the figure legend (e.g., PBD, SHR…), as done in Figure 9.
Table 1: Include the country and city of each company in the “Company” column.
Line 121: For statistical analysis, provide more details—for example, information about data distribution analysis.
Line 125: Verify whether P or p should be italicized.
Line 127: Specify the type of hypertension and revise the subtitle accordingly—for instance, “Effects of a Plant-Based Diet on Lung Tissue of Rats with PH.”
Lines 128–129: The statement “Expression of endothelial nitric oxide synthase (eNOS), a nitric oxide–producing enzyme, and its reduction, which is closely associated with endothelial dysfunction [20–22]” should not be presented as a result since endothelial dysfunction was not directly measured. Consider moving this to the discussion section.
Line 135: If eNOS protein expression was measured in lung tissue, it should not be described as lung microvascular expression. Please revise this throughout the manuscript.
Lines 142–157: Again, since protein expression was measured in lung tissue, it cannot be interpreted as lung epithelial cell integrity and fibrosis. Specify the type of hypertension. Additionally, the initial section reads more like a discussion or conclusion rather than a results section; please revise throughout the results.
Line 161: “After 24 weeks, animals were sacrificed” belongs in the methodology section.
Line 168: Add a space in “(MMP)9…”. This section reads as a discussion; please revise throughout the results section.
Line 170: If α-SMA expression—a protein expressed in epithelial cells during mesenchymal transition and in activated fibroblasts—was measured, please include this in the methods section.
Line 175: “TGF-β1, a protein involved in reducing E-cadherin, mesenchymal transition, and promoting fibrosis,” provides background information, not results.
Line 195: The phrase “animals were sacrificed” belongs in the methodology section.
Line 202: Replace “inflammatory signaling” with “inflammatory protein expression.”
Lines 203–219: Explain why a significant decrease in p-c-Jun expression was observed in SHR+PBD (week 24) but not at week 36.
Line 205: Replace “impacted” with “expressed”; the same applies to line 213.
Line 206: The statement regarding MAPK activity does not constitute a result and should be removed.
Lines 230–247: Please explain whether a decrease in XO and 3-NT levels beyond control conditions (WKY) could have physiological implications for the rats; clarify these findings.
Figure 8: How do you explain the absence of significant changes in lung tissue across all study groups?
Figure 9: Please include the effect of PH in the schematic, replace “pulmonar alveole” with “lung” because western blotting was performed on lung homogenates rather than specific cells, substitute the food image with the specific foods described in the methods section, and remove references to endothelial or epithelial cell stress. Overall, Figure 9 should be revised to align with the actual findings.
Line 321: Clarify why the authors describe the results as “equivocal.”
Lines 324–325: How could eNOS expression in lung tissue homogenates reflect improved lung microvascular endothelial function?
Lines 329–331: Additional studies are needed to substantiate this statement.
Lines 300–364: The entire discussion section requires rewriting. For example, explain whether decreased XO and 3-NT levels beyond control conditions (WKY) could affect rat physiology, clarify the lack of significant changes in lung tissue across all study groups, and discuss how hormonal effects may be associated with your findings. Provide more details about the protein analyses.
Lines 365–368: Strengthen the conclusion to align with the revised discussion and analysis.
Line 388: Remove repeated DOIs, review all references for accuracy, and include more recent studies.
Author Response
We thank the reviewer for their valuable feedback. We’ve endeavored to address their concerns/suggestions to the greatest extent possible. Reviewer comments are in bold while our responses are in plain text.
Line 45: Add a space after “males […]”.
This is complete.
Line 59-64: Please replace the references, since i think that adding different studies improve the strong of the manuscript.
We have added additional references that underscore the role of polyphenols in inflammation, oxidative stress, and endothelial function with the following DOIs to address this: 10.2174/1381612825666190722100504 and 10.1161/ATVBAHA.109.199687
Line 69: Consider adding more information regarding the higher prevalence of PH in women compared to men—for example, age-related differences.
We have added the following text “PH was initially thought to be a disease afflicting mostly younger women (mean age 35); however, recent evidence suggests that it afflicts older adults as well, with 64% of new diagnoses occurring in those >65 years of age [4]”
Line 75–76: Why were the rats doubly housed? Please also provide information regarding the diet, nutrition, and health care of the animal model prior to the experimental protocol, as well as the room temperature and whether all procedures were approved by an ethics committee.
We’ve edited our manuscript to clarify that rats were doubly housed since singly housed rats can result in significant stress, potentially altering the results of the intervention. For example, singly housed rats have been observed to have increased blood pressure. Considering that some of these rats were housed for 36 weeks (~8.2 months), we thought it both unethical and potentially problematic with respect to the animals’ response to treatments. Specifically, we’ve have added the following text.
“Rats were doubly housed since singly animals may have altered hemodynamics due to increased stress [21]. “
We have indicated that “All animal use and procedures were approved by Georgia State University's Institutional Animal Care and Use Committee (protocol #: A23025).”; however, the following housing detail was added “(50 ± 5% relative humidity, 20–25 °C)”. We have also added a new supplementary table (Table S1) which details nutrition information for both diets.
Line 79–83: Why was the concentration of each PBD component set at 4%? Why was it the same for all components, despite their differing polyphenol contents? Finally, what quantity of PBD was administered to each rat, and why?
The 4% value of each plant component (total 28%) was chosen in an effort to reflect the diversity of the human diet, since humans typically do not consume 1 food for most of their caloric needs (e.g., 28% of diet coming from bell pepper). While we did not quantify the polyphenol content due to financial limitations, these foods have been extensively assessed before. We have provided an estimated calculation in the text “Polyphenol intake in rats consuming PBD was estimated to be ~2,582 mg/kg BW based on average food intake, body weight of animals, and available polyphenol analysis data from Phenol Explorer [23,24]. This corresponds to ~96 mg/kg BW of polyphenols in human equivalents [25], or 5,760 mg of polyphenols per 2,000 Kcal.”
Lines 92–93: Add details regarding the methods used for euthanasia.
We have added the following text: “All animals were euthanized by CO2 affixation followed by decapitation”.
Figure 1: Please include explanations for all acronyms in the figure legend (e.g., PBD, SHR…), as done in Figure 9.
All abbreviations for all legends have now been provided.
Table 1: Include the country and city of each company in the “Company” column.
We have included these details in the footer of the table.
Line 121: For statistical analysis, provide more details, for example, information about data distribution analysis.
We have assessed this and included the following text “Normality was assessed with a Shapiro-Wilk test, and all data were normally distributed.”
Line 125: Verify whether P or p should be italicized.
All “P”s for p-values have now been italicized.
Line 127: Specify the type of hypertension and revise the subtitle accordingly—for instance, “Effects of a Plant-Based Diet on Lung Tissue of Rats with PH.”
We appreciate the reviewer’s suggestion on this. However, since we were unable to measure PH directly and only assume that it exists based on existing literature in SHRs, we prefer to take a more conservative approach with our characterization of the effects of the PBD on PH. We plan to assess PH in future SHR studies, however, considering this data.
Lines 128–129: The statement “Expression of endothelial nitric oxide synthase (eNOS), a nitric oxide–producing enzyme, and its reduction, which is closely associated with endothelial dysfunction [20–22]” should not be presented as a result since endothelial dysfunction was not directly measured. Consider moving this to the discussion section.
We appreciate the reviewer’s concern but respectfully suggest that being more conservative in our tone is a less confusing way to address the concern. We’ve edited our text to be careful to not directly claim that endothelial dysfunction is present, only that eNOS closely associates with endothelial dysfunction. “As an indirect means of probing endothelial dysfunction ,we measured expression of endothelial nitric oxide synthase (eNOS), a nitric oxide-producing enzyme whose levels are associated with endothelial dysfunction”, This close association is well described both in the literature (PMID: 25330054) and demonstrated by our group, both in this model (PMID: 40799580) and in human endothelial cells (PMID: 36611888). We believe that this context for presenting eNOS data is important.
Line 135: If eNOS protein expression was measured in lung tissue, it should not be described as lung microvascular expression. Please revise this throughout the manuscript. Lines 324–325: How could eNOS expression in lung tissue homogenates reflect improved lung microvascular endothelial function?
We would like to note that an overwhelming majority of endothelial cells in the lungs are microvascular (DOI: 10.1007/978-3-642-75262-9_9). However, we acknowledge eNOS is expressed in other cell types in the lung as well despite its dominant role in ECs. Thus, we have added this as a limitation, since we cannot be certain of the proportion of eNOS that is derived from endothelial cells. It is important to note that lung ECs express eNOS twice as greatly as in epithelial cells (PMID: 10713142), and that eNOS activity in endothelial cells is found to be reduced in PH (PMID: 27130529 ). Thus, we believe that this eNOS data is of relevance to the microvascular endothelium in the lung. We have included the following: “In addition, we cannot be certain whether the majority of eNOS is from endothelial cells or from other cell types in the lungs which also express eNOS. Prior studies have found that eNOS activity from lung endothelial cells in PH is downregulated—thus, we believe that our data reflects the microvascular endothelium, at least partially [51]. Lung epithelial cells also express eNOS, but this expression is half that of lung endothelial cells [52].”
Lines 142–157: Again, since protein expression was measured in lung tissue, it cannot be interpreted as lung epithelial cell integrity and fibrosis. Specify the type of hypertension.
Thank you for bringing this to our attention. Our tissue cuts were at the end of the lung tissue rather than towards the bronchi; thus, we believe that we have captured substantial alveoli which includes pneumocytes and endothelial cells. Type I pneumocytes comprise a majority of the alveolar epithelium (www.ncbi.nlm.nih.gov/books/NBK534789). Because of this, we believe that our epithelial integrity markers do reflect lung epithelial cells. In fact, E-cadherin staining primarily occurs in lung epithelium rather than surrounding tissue (PMID: 23492370). However, because we have a mixed cell population, we cannot be certain that this reflects epithelial cells exclusively and not also endothelial cells. We have provided this as a limitation. We would like to note that others in the literature have also used lung tissue to assess epithelial cell integrity, from both humans and animals (PMIDs: 23299965, 27982105, 29362432). The following text in the limitation section has been added: “We also are not certain whether our junction markers (Figure 3) reflect purely epithelial cells. However, we aimed to collect lung tissue for protein analysis specifically at the terminal end of the tissue rather than towards the large bronchi; as such, we believe we have captured protein from the end of the bronchial tree which comprises the small airway and contains substantial alveoli. In addition, E-cadherin is observed primarily in epithelial cells of the small airways [53]. As such, we believe that our data primarily reflects epithelial junction proteins rather than junction proteins of other cell types”
Line 161: “After 24 weeks, animals were sacrificed” belongs in the methodology section. Line 195: The phrase “animals were sacrificed” belongs in the methodology section.
We believe providing this information in the legends of the figures helps readers interpret figures without searching for methodological details in the main text. With that said, we also now note it in methods ”All animals were euthanized by CO2 affixation followed by decapitation at either week 24 or week 36.”
Line 168: Add a space in “(MMP)9…”. This section reads as a discussion; please revise throughout the results section. Additionally, the initial section reads more like a discussion or conclusion rather than a results section; please revise throughout the results. Line 175: “TGF-β1, a protein involved in reducing E-cadherin, mesenchymal transition, and promoting fibrosis,” provides background information, not results. Line 206: The statement regarding MAPK activity does not constitute a result and should be removed.
We very much appreciate the reviewer’s feedback and respect their opinion regarding our results section which includes some background. With that said, we claim this is a stylistic preference that we find important to have context for our data as it is presented rather than presented without context to provide more understanding of our data. We understand that background is not always presented in the results section, but this is a matter of preference.
Line 170: If α-SMA expression—a protein expressed in epithelial cells during mesenchymal transition and in activated fibroblasts—was measured, please include this in the methods section.
This protein is included in the antibody list in Table 1.
Line 202: Replace “inflammatory signaling” with “inflammatory protein expression.”
Since we do not show inflammatory cytokines or products of inflammation (COX2, iNOS, etc.), we cannot necessarily claim that these are inflammatory proteins. Rather, these proteins are considered those involved in inflammatory signaling.
Lines 203–219: Explain why a significant decrease in p-c-Jun expression was observed in SHR+PBD (week 24) but not at week 36.
Thank you for this observation. We do not have a clear explanation for this, as the inflammatory process is complex. SAPK/JNK is involved in regulating c-Jun, so this data is at least consistent between weeks 24 and 36. In future studies we plan to take a much more comprehensive approach with regards to our assessment inflammation, perhaps with cytokine arrays or proteomics. Our selection of inflammatory proteins we investigated were limited.
Line 205: Replace “impacted” with “expressed”; the same applies to line 213.
In this context, the word “expressed” would not be appropriate, as we stated “No significant differences in the immune cell infiltration marker, F4/80, were observed between groups at week 24 (Figure 5A, B) or week 36 (Figure 5G, H); nor was the phosphorylation of NF-κB significantly impacted”. Replacing impacted to expressed would not be correct.
Lines 230–247: Please explain whether a decrease in XO and 3-NT levels beyond control conditions (WKY) could have physiological implications for the rats; clarify these findings.
We believe that these redox results are somewhat equivocal and not fully clear. We have added the following text to the discussion regarding this “For example. XO was lower in SHRs at week 24 vs WKY, but not at week 36, while 3-NT was lower in PBD vs SHR at week 24, but not week 36”
Figure 8: How do you explain the absence of significant changes in lung tissue across all study groups?
Previous studies in male SHRs have shown lung abnormalities, but females have not been evaluated before. We have the following limitation to explain this “Lastly, while we did not observe histopathological changes in the lung, the known pathological effects of the changes in the expression of proteins involved in endothelial function, epithelial junctions, and fibrosis were attenuated, suggesting that female SHRs of increased age would likely be needed to show histological abnormalities.”
Figure 9: Please include the effect of PH in the schematic, replace “pulmonar alveole” with “lung” because western blotting was performed on lung homogenates rather than specific cells, substitute the food image with the specific foods described in the methods section, and remove references to endothelial or epithelial cell stress. Overall, Figure 9 should be revised to align with the actual findings.
We appreciate these suggestions by the reviewer. To soften this figure, we have included the word “hypothesized” in the figure title, since we do not have absolute certainty of the mechanisms or specific cell types involved. Also, we do not want to overemphasize PH since we did not confirm its presence in these animals and stating that these effects occurred in essential hypertension is more accurate of our model. Nonetheless, we have changed the foods to reflect what we investigated.
Line 321: Clarify why the authors describe the results as “equivocal.”
We refer to the ambiguity of our findings, since inflammatory signaling/oxidative stress were not clearly or consistently impacted by hypertension or the PBD (aside from XO). Thus, we cannot draw clear conclusions from this data. We have changed the wording of this sentence “Changes in inflammatory signaling and redox proteins were less clear and more ambiguous…”
Lines 329–331: Additional studies are needed to substantiate this statement.
We agree, and the following has been added: “although additional studies are needed”
Lines 300–364: The entire discussion section requires rewriting. For example, explain whether decreased XO and 3-NT levels beyond control conditions (WKY) could affect rat physiology, clarify the lack of significant changes in lung tissue across all study groups, and discuss how hormonal effects may be associated with your findings. Provide more details about the protein analyses.
Because inflammation and oxidative stress data was equivocal, we did not spend significant time discussing this, as we did not have clear directions with this data. Instead, we focused on key data which was consistent between time points. However, we have included additional discussion of the potential role of estrogen.
“Despite the protective role of estrogen in cardiovascular diseases [43], females, including those that are premenopausal, have higher rates of PH than males. Oddly, estrogen is protective in PH in animal models, [44,45], representing a paradox. Thus, it is not clear why females have a higher incidence of PH. Future studies should evaluate sex differences in SHRs and also evaluated young versus aged animals to elucidate these mechanisms.”
Reviewer 2 Report
Comments and Suggestions for Authors
This study explores the effects of a plant-based diet (PBD) on pulmonary hypertension (PH) using a spontaneously hypertensive rat model. Female rats were fed either a refined control diet or a PBD rich in whole fruits, vegetables, legumes, and nuts (28% in total with soy protein replacing casein), with both prevention and treatment arms. The findings demonstrate that PBD preserved and restored endothelial nitric oxide synthase (eNOS), attenuated loss of epithelial integrity markers such as E-cadherin, reduced ERK1/2 phosphorylation, and decreased TGF-β1 activation, all processes implicated in pulmonary vascular stress and fibrosis. The findings suggest that PBD offers potential protective and restorative effects against hypertension-induced pulmonary dysfunction.
The manuscript addresses a novel and original exploration linking nutrition (PBD) to pulmonary vascular health. The diet design was carefully controlled, allowing phytochemicals to be isolated as the main differentiating factor, which added rigor to the interpretation. The breadth of molecular analyses provided a well-rounded mechanistic insight. The inclusion of female rats is commendable, given the sex-specific prevalence of pulmonary hypertension, and enhances translational relevance. The Western Blot figures were clearly presented and well interpreted with strong relevance. Figure 9 is particularly commendable, since it effectively synthesizes the molecular findings into an integrated mechanistic model, making the complex pathways accessible to readers. The discussion was balanced, with clear acknowledgment of limitations and thoughtful connections to human dietary patterns and potential clinical applications.
Please kindly find the following comments/suggestions/questions for improvement:
- What evidence supports that spontaneous hypertension in the rat model had progressed to pulmonary hypertension by week 28 (instead of systemic hypertension)?
- Did you analyze the polyphenol or antioxidant content of the plant-based diet formulation to confirm the phytochemical differences between the PBD and control feed? If there are existing data regarding this, please add.
- Incomplete blot in Figure 6 is deplorable. If possible, the authors may provide a complete blot to replace the figure.
- Please add a Conclusion section and insights for future studies, including how to map the study in humans.
Thank you.
Author Response
We thank the reviewer for their valuable feedback and have aimed to address their concerns/suggestions to the greatest extent possible. Reviewer comments are in bold and our responses are in plain text.
What evidence supports that spontaneous hypertension in the rat model had progressed to pulmonary hypertension by week 28 (instead of systemic hypertension)?
We do not have direct evidence of this unfortunately, and this is provided as a limitation in the discussion. Hence, our title reflects hypertension (rather than PH), and we refer to the proteins we assessed as associated with PH, but never claim that PH is treated in our model. We are hypothesizing that PH is present based on the strain and age of animals based on existing literature (PMID: 8546217), but we plan to assess this directly in future studies, as the present study is a secondary analysis.
Did you analyze the polyphenol or antioxidant content of the plant-based diet formulation to confirm the phytochemical differences between the PBD and control feed? If there are existing data regarding this, please add.
While we did not quantify the polyphenol content due to financial limitations, these foods have been extensively assessed before. We have provided an estimated calculation in the text “Polyphenol intake in rats consuming PBD was estimated to be ~2,582 mg/kg BW based on average food intake, body weight of animals, and available polyphenol analysis data from Phenol Explorer [23,24]. This corresponds to ~96 mg/kg BW of polyphenols in human equivalents [25], or 5,760 mg of polyphenols per 2,000 Kcal.”
Incomplete blot in Figure 6 is deplorable. If possible, the authors may provide a complete blot to replace the figure.
This missing lane was because of a nonspecific band running through the blot. Nonetheless, to address this comment and increase transparency, this band has been placed back and the figure has been updated.
Please add a Conclusion section and insights for future studies, including how to map the study in humans.
We have added a conclusion section and added the following “Pilot clinical studies utilizing PBD to treat human PH may be warranted; however, confirmation of the ability of PBD to target PH itself requires investigation.”
Reviewer 3 Report
Comments and Suggestions for Authors
Dear Authors
I appreciate your efforts to come-up with this draft. All well, little language modifications is required for the points highlighted in attached PDF in comments or suggestions format.
In case, image could be possible to expand, atleast raise little of their pixels or magnification so that they are easy to understand and quote by other researchers of the field.
Wish you Good Luck!!!

scope of improvement is needed to this communication as of the Repo of the Journal is concerned.
Author Response
We thank the reviewer for their valuable feedback and have aimed to address their concerns/suggestions to the greatest extent possible. Reviewer comments are in bold and our responses are in plain text.
it should be before the brackets.
This has been corrected.
arrange these keywords in alphabetical order.
This has been corrected.
Why it is written in that complex format, Control diet disclosure should be so simple in accordance to vegan or animal based constituent inclusion or emission. So, Re-write it.
To clarify the composition of this diet, we have included a supplementary table (Table S1).
Why difference is there among these same color highlighted spellings??
This was a font error which has been corrected.
rewrite it (In reference to equivocal).
We refer to the ambiguity of our findings, since inflammatory signaling/oxidative stress were not clearly or consistently impacted by hypertension or the PBD. Thus, we cannot draw clear conclusions from this data. We have changed the wording of this sentence “Changes in inflammatory signaling and redox proteins were less clear and more ambiguous…”
Rewrite this statement, again confusing and too long sentence so better to write it using technical writing skills.
We have attempted to improve this sentence by dividing it into 2 distinct statements: “However, a clear finding we observed was complete prevention and reversal of diminished eNOS protein expression in lung tissue with PBD supplementation (Figure 2). This suggests that a PBD improves lung microvascular endothelial function independently of hypertension”
Round 2
Reviewer 1 Report
Comments and Suggestions for Authors
The manuscript has improved notably, and each point has been addressed accordingly. However, several aspects still require clarification and correction, as detailed below.
The introduction requires further revision. While the main focus was pulmonary hypertension, the refinement of the study’s aim now demands greater coherence and alignment with the stated objectives.
In the title, please remove the word cellular, as the primary measurements in this study were performed at the molecular level. Please consider reflecting this revision in the stated aim, and I suggest not including references within the aim section.
Line 139: Please remove the term microvascular, since the measurement of eNOS was conducted in lung tissue.
Line 188: The authors have included discussion within the results section; therefore, please move these sentences to the discussion section. Consider applying this suggestion throughout all results segments.
Please include a discussion explaining why TGF-β1 levels decreased in group 24 but not in group 36and please add more discussion about the results obtained.
Line 226: Please remove the word signaling, since this study only measured protein expression through semiquantitative analysis, and did not assess protein interactions to determine inflammatory signaling.
Figure 8: Please check and declare notable in each figure the Magnification in each image.
Line 364-365: please add … PBD could improve…

Author Response
We thank the reviewer for their valuable feedback. We’ve endeavored to address their concerns/suggestions to the greatest extent possible for this second round of revisions. Reviewer comments are in bold while our responses are in plain text.
The manuscript has improved notably, and each point has been addressed accordingly. However, several aspects still require clarification and correction, as detailed below.
The introduction requires further revision. While the main focus was pulmonary hypertension, the refinement of the study’s aim now demands greater coherence and alignment with the stated objectives.
We thank the reviewer for their suggestion regarding rewriting the introduction to be aligned with the study objectives. Upon review, we changed the word “cellular” to “molecular” to better reflect the measures of the study. We also reviewed our conclusions and objectives as they relate to the introduction. We added the phrase “which relate to PH” in the aim to reflect the study scope. Aside from these changes, we believe that the introduction and aims are in tight alignment with each other, and we could not identify incongruence between our objectives and conclusions as it relates to the introduction. For example, we were highly specific, in that we were assessing a model of essential hypertension since we could not confirm that PH was present. We merely describe the molecular abnormalities observed in PH in the 2nd paragraph that are in alignment with what we observed in the present study.
In the title, please remove the word cellular, as the primary measurements in this study were performed at the molecular level. Please consider reflecting this revision in the stated aim, and I suggest not including references within the aim section.
We have altered our title and aims accordingly to state “molecular” and not “cellular”. We did this throughout the manuscript as well. We have also removed the reference in the study objective as advised.
Line 139: Please remove the term microvascular, since the measurement of eNOS was conducted in lung tissue.
We have changed “microvascular endothelium” to “endothelial nitric oxide synthase (eNOS)” for better accuracy.
Line 188: The authors have included discussion within the results section; therefore, please move these sentences to the discussion section. Consider applying this suggestion throughout all results segments.
We very much appreciate the reviewer’s feedback and respect their opinion regarding our results section which includes some background and framing of the data. With that said, we claim this is a stylistic preference that we find important to have context for our data, especially as we transition through our data to provide rationale for its presentation. We believe this context provides more understanding of our results. We understand that this information is not always presented in the results section, but this is a matter of preference and performed frequently in many published papers.
Please include a discussion explaining why TGF-β1 levels decreased in group 24 but not in group 36and please add more discussion about the results obtained.
We would like to apologize for any confusion to the reviewer. TGF-B1 in PBD versus SHR between weeks 24 and 36 were actually similar between groups when the ratio of cleaved/pro TGF-B1 was assessed (Panels F and L). We have added this clarification to the discussion section.
Line 226: Please remove the word signaling, since this study only measured protein expression through semiquantitative analysis, and did not assess protein interactions to determine inflammatory signaling.
We apologize for this confusion. Since the proteins we assessed are part of a signaling cascade and not an end product of inflammation, we believe that referring to these proteins as signaling proteins is more accurate and precise than claiming these are inflammatory proteins. We have added the word “proteins” to reflect that we are assessing inflammatory signaling proteins.
Figure 8: Please check and declare notable in each figure the Magnification in each image.
We have updated the magnification in the legend to state “Total magnification is 200x”.
Line 364-365: please add … PBD could improve…
We have made this change as requested.